# Mediating Effects of Risk Perception on Association between Social Support and Coping with COVID-19: An Online Survey

**DOI:** 10.3390/ijerph18041550

**Published:** 2021-02-06

**Authors:** Dian-Jeng Li, Nai-Ying Ko, Yu-Ping Chang, Cheng-Fang Yen, Yi-Lung Chen

**Affiliations:** 1Graduate Institute of Medicine, College of Medicine, Kaohsiung Medical University, Kaohsiung City 80708, Taiwan; u108800004@kmu.edu.tw; 2Department of Nursing, Meiho University, Pingtung 91202, Taiwan; 3Department of Addiction Science, Kaohsiung Municipal Kai-Syuan Psychiatric Hospital, Kaohsiung City 802211, Taiwan; 4Department of Nursing, College of Medicine, National Cheng Kung University, Tainan City 70140, Taiwan; nyko@mail.ncku.edu.tw; 5School of Nursing, The State University of New York, University at Buffalo, Buffalo, NY 14260, USA; yc73@buffalo.edu; 6Department of Psychiatry, Kaohsiung Medical University Hospital, Kaohsiung City 80708, Taiwan; 7Department of Healthcare Administration, Asia University, Taichung City 41354, Taiwan; 8Department of Psychology, Asia University, Taichung City 41354, Taiwan

**Keywords:** risk perception, confidence, social support, coping strategy, COVID-19, SARS-CoV-2

## Abstract

Coronavirus disease 2019 (COVID-19) is a novel infectious disease which has had a great impact on the public. Further investigations are, therefore, needed to investigate how the public copes with COVID-19. This study aimed to develop a model to estimate the mediating effects of risk perception and confidence on the association between perceived social support and active coping with the COVID-19 pandemic among people in Taiwan. The data of 1970 participants recruited from a Facebook advertisement were analyzed. Perceived social support, active coping with COVID-19, risk perception and confidence were evaluated using self-administered questionnaires. Structural equation modeling was used to verify the direct and indirect effects between variables. The mediation model demonstrated that lower perceived social support was significantly associated with a higher level of active coping with COVID-19, and this was mediated by a higher level of risk perception. The present study identified the importance of risk perception on the public’s coping strategies during the COVID-19 pandemic.

## 1. Introduction

### 1.1. Golobal Scenario of COVID-19 and Coping Strategies for Infective Respiratory Disease Pandemics

Coronavirus disease 2019 (COVID-19) is a novel infectious respiratory disease caused by a novel coronavirus (SARS-CoV-2). It causes physical symptoms including severe pneumonia, pulmonary edema and multiple organ failure [1]. It emerged at the end of 2019 and soon became a major public health burden worldwide [2]. On 11 March 2020, the WHO declared the COVID-19 outbreak a pandemic due to the rapid virus spread and high death toll over the world [3]. As of 29 January 2021, there were up to one hundred million confirmed cases and over two million deaths in over 200 countries [4].

People may change their daily routines due to the adoption of protective behaviors against COVID-19 and search for additional information on the disease. Understanding how the public cope with a pandemic can help health professionals better understand the impact it has on their daily lives, the adequacy of policy for infection control, and the future outcomes of the pandemic. For instance, handwashing is the behavior most recommended by the World Health Organization to protect individuals from contracting COVID-19 [5]. Several coping strategies during infectious disease pandemic were frequently used, such as active coping (seeking social support), problem-focused coping (seeking alternatives, problem-solving), and emotion-focused coping (avoidance) [6]. A longitudinal study recruiting publics during COVID-19 also indicated that several coping strategies, specifically seeking social support, engaging in distractions, and seeking professional help, were used more frequently by those with more pandemic/lockdown distress [7].

In contrast with negative/passive coping, active coping is a stress-management strategy in which a person directly works to control a stressor through targeted behavior [8]. It is generally considered adaptive, having been associated with fewer mood disturbances, and enhanced self-efficacy [8]. Different types of coping strategies are associated with diversities of psychological impacts. During the severe acute respiratory syndrome (SARS) pandemic, active coping was positively related to perceived general health and life satisfaction [9]. It was also reported to be associated with positively subjective wellbeing in COVID-19 pandemic [10]. On the other hand, a web-based survey of people in China reported that those with negative/passive coping strategies, such as do nothing or substance abuse, had a higher level of psychological distress during the COVID-19 epidemic [11]. Moreover, individuals who have negative coping strategies for the COVID-19 pandemic have a higher risk of being infected. For example, people with cognitive impairment and mental illness are more vulnerable to COVID-19 infection as they have little awareness of the risk and maladaptive coping strategies regarding personal protection [12]. Therefore, investigations into factors that predict how the public actively cope with the COVID-19 pandemic are crucial to estimate the multi-dimensional impacts of COVID-19.

### 1.2. Influence of Perceived Social Support, Risk Perception, and Confidence with Active Coping

Whether perceived social support affects individuals’ coping strategies against the threat of COVID-19 remains unclear. Chao reported that higher social support was positively associated with problem-focused coping among the elderly who experienced Typhoon Morakot in Taiwan [13]. In addition, a study in the US revealed that support via financial security was a predictor of adherence to the Centers for Disease Control and Prevention (CDC) guidelines for infection control of COVID-19 [14]. However, how perceived social support influences coping strategies against COVID-19 is not clear. Therefore, further studies are needed to investigate whether there are factors that mediate the association between perceived social support and active coping with the COVID-19 pandemic.

A meta-analysis of experimental studies demonstrated that people’s intentions and behavior change following heightened risk appraisal, including risk perception [15]. Several psychological or social factors are reported to be associated with risk perceptions of COVID-19. Improving perceptions about infectious diseases in society could lead to a significant improvement in a patient’s well-being and decrease in discrimination [16]. In addition, prosocial values, trust in government, science, and medical professionals, and personal knowledge of COVID-19 were all significant predictors of risk perception [17]. Estimating the level of risk perception may be important for the public because that it will affect the public’s behaviors or coping with COVID-19. It was reported that social distancing and hand washing were strongly predicted by the perceived probability of personally being infected, which is a kind of risk perception [18]. Another cross-sectional study in Mexico demonstrated that both higher level of perceived susceptibility and perceived severity of COVID-19 were associated with protective behaviors of staying home [19]. On the other hand, confidence in coping with the COVID-19 pandemic may be associated with active coping with COVID-19. Confidence in coping is similar to self-efficacy, representing the individuals’ beliefs that they have the ability to do specific tasks in the future [20]. Previous studies have reported significant associations between having more knowledge about disease and self-efficacy in coping with SARS [21] along with COVID-19 [22]. Therefore, further studies are warranted to investigate whether risk perception and confidence mediate the association between social support and active coping with COVID-19.

### 1.3. Aims of the Current Study

Adopting adequate coping strategies for infective respiratory disease pandemics affects both personal health and also the efficacy of infection control for society as a whole. The aims of the present study were to identify any associations between perceived social support and active coping with the COVID-19 pandemic, and the potentially mediating effects of risk perception and confidence. According to above reviews of literatures, it is hypothesized that perceived social support may be associated with active coping with the COVID-19 pandemic. Moreover, either confidence or risk perception may be partial or full mediated the association between perceived social support and active coping with COVID-19.

## 2. Methods

### 2.1. Participants and Procedures

The current study was based on dataset of the Survey of Health Behaviors During the COVID-19 Pandemic in Taiwan, which was initially reported elsewhere [22]. The expert meeting was held to develop questionnaires, which were used in this study. In brief, Facebook users aged ≥20 years and living in Taiwan were recruited into this study between 10 April and 23 April 2020. A Facebook advertisement was posted, which included a headline, main text, pop-up banner and weblink to the research questionnaire website. The recruiting advertisement was designed to appear in the “News Feed” of Facebook, which is a streaming list of updates from the user’s connections (e.g., friends) and advertisers. A previous study indicated that News Feed advertisements are more effective in terms of recruitment metrics for research studies [23]. In order to increase its visibility, we also posted the online advertisement to Line and Facebook groups.

This study was approved by the Institutional Review Board of Kaohsiung Medical University Hospital (approval no. KMUHIRB-EXEMPT(I)20200011). Although the participants were not given any incentive for their participation, at the end of the questionnaire we provided them with weblinks to the online COVID-19 Information Centers of the Taiwanese CDC, Kaohsiung Medical University Hospital, and the Medical College of National Cheng Kung University so they could search for useful information.

### 2.2. Questionnaires

#### 2.2.1. Perceived Social Support

We estimated the levels of satisfaction with perceived social support using three questions: “In the past week, did you receive satisfactory support from your (1) family, (2) friends, and (3) colleagues or classmates?” The responses were graded on a five-point Likert scale, with scores ranging from 0 (entirely disappointed) to 4 (extremely satisfied). Higher total scores indicated more satisfaction with their level of perceived social support during the COVID-19 pandemic. This instrument is reliable and well-validated according to the supplementary material of previous publication [24].

#### 2.2.2. Active Coping with COVID-19

Liao et al. [25] developed several questionnaires to estimate the protective behavior in the 2009 influenza A/H1N1 pandemic in Hong Kong. In reference to the above study, we developed 7 questions to assess the respondents’ level of active coping with the threat of COVID-19 during their daily lives [26]. The active coping with COVID-19 represented the coping strategies of problem solving (protective behaviors) against the threat of COVID-19. These questions asked participants if they: (1) avoided going to crowded places, (2) maintained good indoor ventilation, (3) cleaned or disinfected their house more often, (4) washed their hands more often, (5) wore a mask, (6) searched for information on COVID-19, and (7) avoided clinic visits or had missed appointments at clinics in the past week. The responses were scored as 0 (“no” or “yes, but not due to COVID-19”) and 1 (“yes, due to COVID-19”). 

#### 2.2.3. Risk Perception toward COVID-19

According to Liao et al. [25], we developed the following question to assess the severity of current worry towards COVID-19: “Please rate your level of current worry towards COVID-19.” The severity of current worry towards COVID-19 was rated from 1 (minimal) to 10 (extremely severe). We also developed four additional questions to evaluate different categories of risk perception: (1) “If you developed flu-like symptoms tomorrow, would you be worried? Reply: 1 (not at all) to 5 (extremely)”, (2) “In the past week, have you worried about catching COVID-19? Reply: 1 (not at all) to 5 (extremely)”, (3) “How likely do you think it is that you will contract COVID-19 over the next month? Reply: 1 (impossible) to 7 (guaranteed)”, and (4) “What do you think your chances are of getting COVID-19 over the next month compared with others outside your family? Reply: 1 (impossible) to 7 (guaranteed)”. The current measurement is reported to be reliable and well-validated according to the supplementary material of previous publication [24].

#### 2.2.4. Confidence against COVID-19

Self-confidence about COVID-19 and perceived confidence in the local government’s ability to control the COVID-19 pandemic were assessed using the following 2 questions: (1) “How confident are you that you will overcome the threats of the COVID-19 pandemic?” and (2) “How confident are you that your city is controlling the COVID-19 pandemic?” The responses were scored on a five-point Likert scale as follows: 0 (not at all confident), 1 (not very confident), 2 (neutral), 3 (confident), and 4 (very confident). Higher scores indicated that the individual was more confident about overcoming the COVID-19 pandemic.

#### 2.2.5. Statistical Analysis

To examine the hypothesized multiple mediation model for the association between perceived social support and active coping with COVID-19, which was mediated by risk perception or confidence (Figure 1), the following analyses were conducted using SPSS and AMOS version 23.0 for Windows (SPSS Inc., Chicago, IL, USA). We examined bivariate associations among the variables using Pearson’s correlation coefficient (*r*), followed by two steps of structural equation modeling (SEM). First, confirmatory factor analysis (CFA) was used to verify the association between latent variables and their indicators in the measurement model. Each question was composed of observed variables (indicators) and latent variables, which indicated perceived social support, active coping with COVID-19, risk perception, and confidence. Factor loading was used as an index to assess the scale reliability between indicators and the corresponding latent variables in the CFA. In addition, Cronbach’s α was calculated to examine the internal consistency reliability. The range was considered acceptable if Cronbach’s α was >0.5 [27]. To estimate the sample adequacy of “active coping with COVID-19” in factor analysis, the Kaiser–Mayer–Olkin (KMO) measure of sampling adequacy and Bartlett testing were applied. A KMO value of >0.60 and statistically significant value of *p* < 0.05 from Bartlett testing indicated the data was adequate for factor analysis [28]. Then, the total variance explained (%) was also estimated through EFA to estimate the validity of “active coping with COVID-19”.

Latent variable path analysis with maximum likelihood parameter estimations was used to estimate the model adequacy and the direct/indirect effects of perceived social support on active coping with COVID-19 through risk perception or confidence [29]. Bootstrapping method with 5000 samples was applied in the path analysis due to the non-normality of the data (Kolmogorov-Smirnov test; *p* < 0.001). As a multiple mediator model, both mediators were applied into the model to assess and compare the mediating effects. As there was a relatively high proportion of females in the study cohort and as the Kolmogorov-Smirnov test (*p* < 0.001) for age was significant, indicating non-normal distribution, age and gender were also included in the multiple mediators’ model as covariates to adjust for their effects on the latent variables. Gender (female, male and transgender) was transformed into two dichotomous dummy variables (male vs. female; and transgender vs. female) for the analysis. The standardized estimates (beta coefficient; β) were reported for the predictive strength explained in the model.

We used the Sobel test to verify the mediating effect [30]. Furthermore, to test the adequacy of the model, multiple indices were applied to verify the goodness of fit. For each of these fit indices, the values indicating an acceptable model fit were as follows: Goodness of Fit Index (GFI ≥ 0.9); Adjusted Goodness of Fit Index (AGFI ≥ 0.9); root mean square error of approximation (RMSEA < 0.08); and standardized root mean square residual (SRMR ≤ 0.08) [31,32].

## 3. Results

### 3.1. Descriptive Statistics, Factor Analysis, and the Correlation Matrix

Initially, 2031 respondents filled in the online questionnaire. After excluding those with missing values (*n* = 31) and those aged below 20 years (*n* = 30), a total of 1970 participants (1305 females, 650 males, and 15 transgender) were included in the analysis. The mean age of the participants was 37.81 ± 11.00 years. The correlation matrix with significance, mean and standard deviation for each indicator is shown in Table 1. In general, active coping with COVIDD-19 is postively correlated with risk perception, but negatively correlated with perceived social support. The values of Cronbach’s α of all questionnares were above 0.5, indicating acceptable range [27]. Regarding the EFA of “active coping with COVID-19”, the value of the KMO coefficient was 0.70, and the Bartletts’ test of sphericity reached statistical significance (*p* < 0.01). It supported the adequeacy of the sample. The total variance explained (%) of “active coping with COVID-19” was at 43.29%, which was close to the acceptable range of 50% [33].

### 3.2. Tests for the Mediation Model and Estimated Coefficient Paths

The first step of the SEM estimated the factor loadings through CFA (Table 2). The results of the reliability test are also presented, which indicated an acceptable range of reliability. After adjusting for age and gender, the multiple mediator model was used to estimate the indirect and direct effects, and the estimated path coefficients are illustrated in Figure 2. We found that an indirect effect at a value of −0.06 reached statistical significance (Sobel test: Z = −4.05; *p* < 0.05), and this was based on the product terms of the path from perceived social support to risk perception (β = −0.13, *p* < 0.001) and the path from risk perception to active coping with COVID-19 (β = 0.49, *p* < 0.001). On the other hand, the mediating effect of confidence on the path between perceived social support and active coping with COVID-19 was not significant (Sobel test: Z = 0.99; *p* = 0.32). Moreover, the direct effect from perceived social support to active coping with COVID-19 was not statistically significant. The significance of the path analysis did not change after adjusting for age and gender. 

These results confirmed the mediating effect of risk perception on the association between perceived social support and active coping with COVID-19. Based on the model fit index, the hypothesized model had an adequate model fit index for RMESA (0.068), GFI (0.927), AGFI (0.902), and SRMR (0.069), indicating the good fit of our hypothesized mediation model.

## 4. Discussion

### 4.1. Main Findings of the Current Study

In the current study, an indirect effect was found in that lower perceived support was significantly associated with a higher level of coping with COVID-19, which was mediated by a higher level of risk perception. In addition, a direct effect of perceived social support on coping with COVID-19 and another indirect effect mediated by confidence against COVID-19 did not reach statistical significance. 

### 4.2. Mediating Effect of Risk Perception on the Association between Perceived Social Support and Active Coping with COVID-19

A higher level of risk perception fully mediated the association between lower perceived support and a higher level of active coping with COVID-19. Although a previous study indicated that financial security predicted better coping strategies against COVID-19 [14], the association between perceived social support and active coping with COVID-19 may be different. Perceived social support represents satisfaction with the general support provided by family, friends, and colleagues/classmates, and this represents broader domains than financial support. In addition, although it did not investigate infective respiratory diseases, a previous study demonstrated that a higher level of social support was associated with a lower perceived risk of breast cancer [34]. O’Sullivan reported that individuals with a higher level of perceived social support may feel that they are relatively safe, leading to optimism bias, which causes them to believe that they are less likely to experience negative events [35]. Individuals with such bias may underestimate their risk of COVID-19; however, further studies are needed to test the effects of optimism bias on risk perception.

In the current study, we found that a higher level of risk perception was associated with a higher level of active coping with COVID-19. A previous study investigated the association between risk perception and coping strategies in patients with diabetes, and found that those who had a low premorbid perception of risks often engaged in diabetes-related risky behaviors [36]. In addition, a systematic review demonstrated that healthcare workers’ risk perception influenced their behavior towards patients and facilitated risk-mitigating strategies for emerging acute respiratory infection diseases [37]. Further prospective studies may provide a better understanding of the temporal relationship between risk perception and active coping in relation to infective respiratory diseases.

The above findings revealed the importance of risk perception on active coping with COVID-19; however, perceived social support can compromise the level of risk perception, leading the interference in active coping with COVID-19. It manifested the controversial role of perceived social support. Previous study reported that higher level of perceived social support was associated with less sleep disturbance and suicidal thought, indicating the protective effect of perceived social support from mental burden [26]. It implicated that interventions in risk perception and perceived social support are both important for publics during the COVID-19 pandemic. Specific support to facilitate social interaction is crucial for those who are socially isolated or quarantined due to infection. Telecommunication or online gathering should also be promoted for the time in need of social distancing. Whereas, intervention to enhance publics’ risk perception should not be neglected. Medical information, news, and governmental policies regarding COVID-19 pandemic should also be announced widely to enhance the risk perception of publics [17].

### 4.3. The Non-Significant Mediating Effect of Confidence on the Association between Perceived Social Support and Active Coping with COVID-19

We found that perceived social support was positively associated with confidence, whereas the association between confidence and coping with COVID-19 was not significant. A cross-sectional observational study on medical staff treating patients with COVID-19 in China demonstrated that levels of social support were significantly associated with self-efficacy [38]. Self-efficacy represents how well one can execute courses of action required to deal with prospective situations, and indicates an individual’s belief that they can overcome obstacles [39]. Although confidence against COVID-19 cannot be entirely compared with self-efficacy, the association between perceived social support and confidence observed in the current study deserves further investigation to explore the potential effect of social support on self-efficacy. 

On the other hand, the insignificant association between confidence and active coping with COVID-19 means that confidence failed to significantly mediate the association between perceived social support and active coping with COVID-19. Since previous studies have emphasized the significant association between gathering information and confidence [22,40], gathering information was only considered as part of active coping with COVID-19 in the current study. This unexpected finding violated the hypothesis of the current study. Several factors may implicate this insignificant association. First, it is possible that other factors involved in active coping with COVID-19 interfered with the association. On the other hand, the questionnaires of confidence in the current study may be insufficient to entirely measure the self-efficacy of participants. Therefore, further development of conceptual model with comprehensively psychological factors and detailed questionnaires measuring self-efficacy may be helpful to determine the detailed interactions between confidence and coping strategies against COVID-19.

### 4.4. Limitations

There are several limitations to the present study. First, possible selection bias may have confounded the results, as the participants were only recruited through a Facebook advertisement. Second, causality could only be inferred among the variables due to the cross-sectional design of this study. Third, several measurements which were crucial in this scenario were not estimated in the questionnaires, such as level of stigma [16], psychological distress, and symptoms of post-traumatic stress disorder (PTSD). Finally, COVID-19 had a limited impact in Taiwan in comparison with other countries, so whether our results can be generalized to other countries is unclear and warrants further investigation.

## 5. Conclusions

In the present study, we found that lower perceived social support was indirectly associated with increased active coping against COVID-19, and that this association was mediated by higher risk perception. However, we did not identify a mediating effect of confidence or a direct effect between perceived social support and active coping with COVID-19. The implication of the current study is that intervention to enhance both perceived social support and risk perception are necessary for public during COVID-19 pandemic. Moreover, risk perception could be more effective to enhance active coping with COVID-19 than the confidence against COVID-19. The subjects who were satisfied with their social support might have had optimism bias that weakened their risk perception and had a compromising effect on their motivation to cope with COVID-19. Since the inversed association between perceived social support and risk perception, it is critical to reduce the effect of optimism bias resulting from perceived social support but not reduce the social support. To enhance perceived social support, specific resources to facilitate social interaction are warranted under adequate infection control. Regarding the impact of the problematic internet use, it is still necessary to promote the telecommunication, online gathering, or programs of social interaction at the difficult time of social distancing. In order to strengthen the risk perception and weaken the effect of optimism bias, facilitation of individuals’ recognition to this pandemic may be beneficial. Timely and correct information about current threats, policies, and strategies against COVID-19 are necessary and should be announced by the authorities through traditional (newspapers or television news) and digital media, such as news feed or livestream thought social software. Public education on infection control is also necessary both during infectious disease outbreaks and at other times. 

We have several suggestions for further research, which could help extend the findings of the present study. A paper-and-pencil questionnaire as opposed to a digital questionnaire, along with printed advertisements posted in public areas would be beneficial to also include non-netizens in the study population. Additional psycho-social factors should also be considered, such as stigma, discrimination, psychological distress, and symptoms of post-traumatic stress disorder. Moreover, further studies investigating optimism bias and self-efficacy using the General Self-Efficacy Scale [41] may be helpful to explore how people cope with the threats of COVID-19. Finally, the prospective cohort study estimating the self-efficacy, risk perception, coping with COVID-19, perceived support and related psycho-social factors (stigma, discrimination, symptoms of PTSD, psychological distress, vaccine hesitancy, etc.) at different stages of this pandemic are warranted. Importantly, attitude or hesitancy of vaccine may be associated with risk perception or coping with COVID-19. Measurements at different stages will be helpful to verify the conceptual model.

## Figures and Tables

**Figure 1 ijerph-18-01550-f001:**
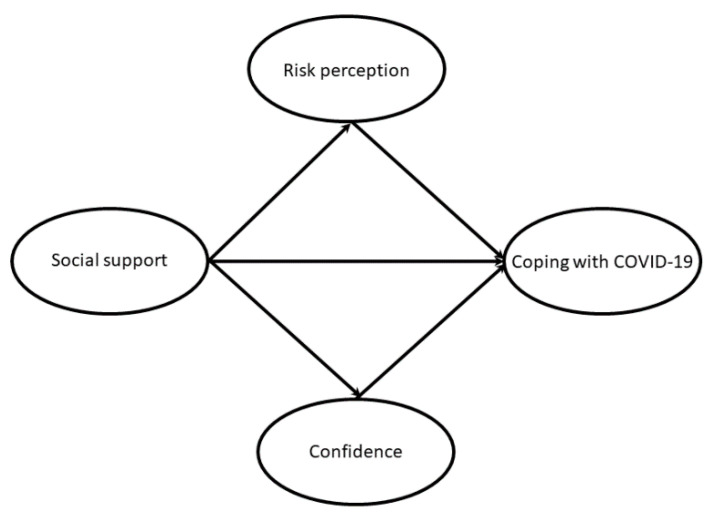
The conceptual model of mediating effect.

**Figure 2 ijerph-18-01550-f002:**
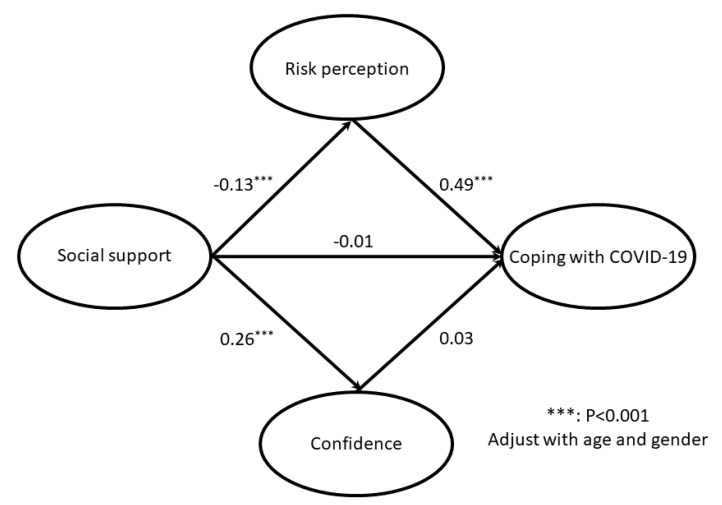
Final model of mediating effect indicating the estimated coefficients of the paths.

**Table 1 ijerph-18-01550-t001:** The correlation matrix of observed variables.

Varaible	Mean	SD	2	3	4	5	6	7	8	9	10	11	12	13	14	15	16	17
1	0.94	0.23	0.13 *	0.14 *	0.16 *	0.18 *	0.09 *	0.06 *	0.18 *	0.1 *	0.18 *	0.11 *	0.04	−0.1 *	−0.05 *	0.01	<0.01	−0.01
2	0.88	0.32	-	0.24 *	0.15 *	0.12 *	0.14 *	0.09 *	0.05 *	0.05 *	0.12 *	0.04	<0.01	−0.02	0.01	−0.01	−0.02	−0.001
3	0.67	0.47		-	0.27 *	0.19 *	0.22 *	0.15 *	0.16 *	0.21 *	0.23 *	0.09 *	0.04	−0.09 *	−0.07 *	−0.02	−0.04	−0.06 *
4	0.92	0.28			-	0.39 *	0.18 *	0.07 *	0.16 *	0.15 *	0.16 *	0.08 *	0.07 *	−0.05 *	−0.04	0.03	−0.01	−0.02
5	0.89	0.31				-	0.21 *	0.1 *	0.18 *	0.18 *	0.22 *	0.13 *	0.12 *	−0.07 *	−0.04	0.02	−0.004	−0.02
6	0.76	0.43					-	0.12 *	0.18 *	0.17 *	0.21 *	0.11 *	0.05 *	−0.9 *	−0.05 *	−0.04	−0.05 *	−0.004
7	0.17	0.37						-	0.07 *	0.13 *	0.09 *	0.08 *	0.03	−0.1 *	−0.04	−0.05 *	−0.07 *	−0.06 *
8	3.93	0.92							-	0.45 *	0.48 *	0.27 *	0.18 *	−0.22 *	−0.13 *	0.01	−0.02	<0.01
9	2.59	0.99								-	0.55 *	0.46 *	0.33 *	−0.31 *	−0.23 *	−0.06 *	−0.1 *	−0.1 *
10	6.14	2.25									-	0.37 *	0.23 *	−0.32 *	−0.24 *	−0.02	−0.05 *	−0.06 *
11	3.47	1.14										-	0.57 *	−0.39 *	−0.27 *	−0.09 *	−0.09 *	−0.09 *
12	3.53	1.28											-	−0.23 *	−0.17 *	−0.04	−0.02	−0.05 *
13	2.41	0.84												-	−0.54 *	0.16 *	0.18 *	0.18 *
14	2.32	0.95													-	0.12 *	0.13 *	0.17 *
15	2.98	0.80														-	0.62 *	0.51 *
16	2.90	0.72															-	0.67 *
17	2.71	0.83																-

*: *p* < 0.05; 1 = Coping-1; 2 = Coping-2; 3 = Coping-3; 4 = Coping-4; 5 = Coping-5; 6 = Coping-6; 7 = Coping-7; 8 = Risk-1; 9 = Risk-2; 10 = Risk-3; 11 = Risk-4; 12 = Risk-5; 13 = Con-1; 14 = Con-2; 15 = Support-1; 16 = Support-2; 17 = Support-3; details of abbreviations are listed in Table 2.

**Table 2 ijerph-18-01550-t002:** Principle component analysis for factors in the conceptual model.

Latent Variables/Observed Variables	Factor Loading	Cronbach’s Alpha
Active coping with COIVD-19		0.56
Avoid going to crowded places (Coping-1)	0.32	
Keep good indoor ventilation (Coping-2)	0.33	
Disinfect house more often (Coping-3)	0.50	
Wash hands more often (Coping-4)	0.55	
Wear a mask (Coping-5)	0.53	
Search information of COVID-19 (Coping-6)	0.40	
Prevent clinic visits or lost follow up (Coping-7)	0.23	
Risk perception		0.71
Develop flu-like symptoms tomorrow (Risk-1)	0.57	
Worried about catching COVID-19 last week (Risk-2)	0.76	
Rate current level of your worry to COVID-19 (Risk-3)	0.71	
How likely you will contract COVID-19 (Risk-4)	0.61	
Chances of getting COVID-19 next 1 month (Risk-5)	0.46	
Confidence against COVID-19		0.70
Self-confidence overcoming threats of COVID-19 (Con-1)	0.89	
Perceived confidence of regional government (Con-2)	0.61	
Perceived social support		0.81
Family members (Support-1)	0.69	
Friends (Support-2)	0.89	
Colleagues or classmates (Support-3)	0.75	

## Data Availability

The data presented in this study are available on request from the corresponding author.

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
