# Peer review of "Mediating Effects of Risk Perception on Association between Social Support and Coping with COVID-19: An Online Survey"

_ijerph, 2021, doi:10.3390/ijerph18041550_

Round 1

Reviewer 1 Report

The present study is an investigation of the association between perceived social support and active coping with the COVID-19 pandemic in a large sample of people in Taiwan. Unfortunately, I have several concerns about this paper. In my opinion, main issues are the lack of a theoretical framework to support the authors' path model, along with the unreliability of measures and inadequate testing of mediation.

Introduction

  1. The authors refer to "active coping strategies" and "negative coping strategies" without providing any definition. The introduction section should be expanded in order to help readers understanding what they refer to. Indeed, there is a wide literature about coping and its relationship with mental health and psychological distress and it may be useful being more explicit about the classification (e.g., dispositional Vs. situational coping; problem-focused Vs. emotion-focused coping) they took into account in their study.
  2. Headings for paragraphs 1.2. and 1.3 are the same. Is this a typo? Anyway, I suggest merging the content of these two paragraphs since they focus on highly related topics. Moreover, some recent studies that considered the role of coping strategies in the context of the COVID-19 pandemic have to be mentioned. See for example:
    • Chew, Q. H., Wei, K.C., Vasoo, S., Chua, H.C. & Sim, K. (2020). Narrative synthesis of psychological and coping responses towards emerging infectious disease outbreaks in the general population: practical considerations for the COVID-19 pandemic. Singapore medical journal, 61(7), 350–356. https://doi.org/10.11622/smedj. 2020046
    • Shanahan, L., Steinhoff, A., Bechtiger, L., Murray, A. L., Nivette, A., Hepp, U., Ribeaud, D., & Eisner, M. (2020). Emotional distress in young adults during the COVID-19 pandemic: evidence of risk and resilience from a longitudinal cohort study. Psychological medicine, 1–10. Advance online publication. https://doi.org/10.1017/S003329172000241X
    • Zacher, H., & Rudolph, C. W. (2020). Individual differences and changes in subjective wellbeing during the early stages of the COVID-19 pandemic. American Psychologist. Advance online publication. http://dx.doi.org/10.1037/amp0000702
  1. The concept of "confidence in coping with the COVID-19 pandemic" is unclear to me. I suggest the authors to better explain what they are talking about.
  2. The authors do no report specific hypotheses. However, in light of previous literature, they can reasonably identify specific predictions. I recommend listing some hypotheses in the final paragraph of the introduction section.

Methods

  1. In general, the study relies on unvalidated self-report measures, which raises questions about the reliability and validity of results. An important issue pertains the internal consistency value of the "Active coping" variable, which is insufficient.
  2. The “Active coping” measure appears to assess the endorsement of protective behaviors rather than active coping. Again, there is a lack of clarity in the definition of the construct under investigation. Moreover, if I properly understand, items were developed in the context of the 2009 influenza A/H1N1 pandemic in Hong Kong (and not COVID-19, as reported).
  3. The authors used the Sobel test. But when computing mediating/indirect effects, the Sobel test's computation of indirect effect standard errors renders these standard errors to not be normally distributed on a sampling distribution. Bootstrapping would need to be done in order to achieve accurate standard errors for the indirect effects.               

Results

Table 1 lacks a legend, which makes impossibile interpreting the correlation matrix. Moreover, the authors do not provide any description of findings reported into this table.

Discussion

The Discussion does not provide a clear and exhaustive interpretation of emerged findings.

Author Response

as attached file

Reviewer 2 Report

Dear Authors,

the impact of the current pandemic is transversal to human nature itself, affecting health and well-being at all levels.

Your contribution proposal needs a finishing touch and some insights in order to deserve publication, since it does not reach sufficient level.

  1. please contextualize the pandemic scenario globally;
  2. please better expiation the concept of risk perception and which factor could influence it;
  3. paragraphs 1.2 and 1.3 have same title;
  4. why not SARS-CoV-2 among keywords?
  5. the questionnaire administered is tailor-made, right? If so, does it contain parts of validated questionnaires by far? Please provide detailed informations;
  6. in such a peculiar scenario you should consider other factors: what about risk stress, PTSD, burnout, stigma and discrimination, FOMO?! You have to deal with psi-factor;
  7. conclusion must be improved. What are the possible repercussions? What suggestions to give to the health policy maker? Define a clear "take home message" from your perspective and address a conclusion section. You need conclusions;
  8. Please state in the conclusion if you will re-contact participants to retake the questionnaire after the pandemic or after they get vaccine;
  9. what about vaccine hesitancy in this scenario? deal with it, even because CoViD-19 vaccines are now available all over the globe (please also refer to https://www.who.int/news-room/spotlight/ten-threats-to-global-health-in-2019 ) and it's crucial in risk perception

You need to significantly improve the manuscript in order to deserve publication.

Please update these gaps referring to the following references:

  • Irigoyen-Camacho, M.E.; Velazquez-Alva, M.C.; Zepeda-Zepeda, M.A.; Cabrer-Rosales, M.F.; Lazarevich, I.; Castaño-Seiquer, A. Effect of Income Level and Perception of Susceptibility and Severity of COVID-19 on Stay-at-Home Preventive Behavior in a Group of Older Adults in Mexico City. Int. J. Environ. Res. Public Health 2020, 17, 7418
  • Baldassarre, A.; Giorgi, G.; Alessio, F.; Lulli, L.G.; Arcangeli, G.; Mucci, N. Stigma and Discrimination (SAD) at the Time of the SARS-CoV-2 Pandemic. Int. J. Environ. Res. Public Health 2020, 17, 6341
  • Sarah Dryhurst, Claudia R. Schneider, John Kerr, Alexandra L. J. Freeman, Gabriel Recchia, Anne Marthe van der Bles, David Spiegelhalter & Sander van der Linden (2020) Risk perceptions of COVID-19 around the world, Journal of Risk Research, DOI: 10.1080/13669877.2020.1758193
  • Wise, T., et al. (2020) Changes in risk perception and self-reported protective behaviour during the first week of the COVID-19 pandemic in the United States. Royal Society Open Science. doi.org/10.1098/rsos.200742

Author Response

as attached file

Round 2

Reviewer 1 Report

I believe that the Authors have done a good job in their revision and the overall quality ho the manuscript has much improved. 

Reviewer 2 Report

Dear Authors,

thank for addressing reviewers' suggestions, resulting in an overall improvement of your contribution proposal.

Best regards